# Leptin Upregulates the Expression of β3-Integrin, MMP9, HB-EGF, and IL-1β in Primary Porcine Endometrium Epithelial Cells In Vitro

**DOI:** 10.3390/ijerph17186508

**Published:** 2020-09-07

**Authors:** Hongfang Wang, Jinlian Fu, Aiguo Wang

**Affiliations:** 1College of Animal Science and Technology, China Agricultural University, Beijing 100193, China; whf@sdau.edu.cn; 2College of Animal Science and Technology, Shandong Agricultural University, Tai’an 271018, China

**Keywords:** leptin, porcine, endometrial epithelial cell, implantation

## Abstract

Obesity has become a global health problem. Research suggests that leptin, a hormone that responds to fat deposition, may be involved in mammalian reproduction; however, its precise role in embryo implantation is poorly understood. Here, primary porcine endometrium epithelium cells (PEECs) were cultured in vitro and used to evaluate the regulatory role of different leptin levels on β3-integrin, MMP9, HB-EGF, and IL-1β, which are, respectively, involved in four critical steps of embryo implantation. Results showed that only 0.01 nM leptin significantly improved β3-integrin mRNA expression (*p* < 0.05). MMP9 and HB-EGF mRNA expressions were upregulated by 0.10–10.00 nM leptin (*p* < 0.05). The IL-1β expression level was only increased by 10.00 nM leptin (*p* < 0.05). β3-integrin, MMP9, HB-EGF, and IL-1β mRNA and protein have a similar fluctuant response to increased leptin. Leptin’s influence on β3-integrin, MMP9, HB-EGF, and IL-1β disappeared when the JAK2, PI(3)K, or MAPK signaling pathways were blocked, respectively. In conclusion, leptin affected porcine implantation by regulating the expression of β3-integrin, MMP9, HB-EGF, and IL-1β in a dose-dependent manner. The signaling pathways of JAK2, PI(3)K, and MAPK may participate in this regulatory process. These findings will contribute to further understanding the mechanisms of reproductive disorders in obesity.

## 1. Introduction

Obesity is a global health problem and its negative effects on reproduction have recently been highlighted. Leptin, which has a similar action in obesity to that of insulin in diabetes, is a hormone that responds to fat deposition. It participates in the regulation of sugar, fat, and energy metabolism; reduces nutrient intake; increases energy release; and inhibits the synthesis of fat, thereby increasing weight loss. In obese females, there are two extremes of leptin levels. The first is *ob* gene mutation, which causes insufficient leptin, and the second is resistance to leptin that forces the *ob* gene to be overexpressed. However, the *ob* gene mutation causing leptin deficiency is rare in the vast majority of obese people, but *ob* gene overexpression inducing supra physiologi levels of leptin is common [1]. In addition to its role in the regulation of food intake and energy balance, leptin also influences mammalian reproduction [2]. The exogenous administration of leptin can rescue sterility in leptin-defective mice (ob/ob mice) [3], which demonstrates its indispensable reproductive role. In our previous study, both leptin and its long-form receptor were shown to be overexpressed at implantation sites compared with inter-implantation sites in porcine endometrium, which suggests that leptin may be a potential regulatory factor in porcine implantation [4]. Despite existing evidence demonstrating leptin’s involvement in pre-implantation embryo development and implantation in humans, mice, and pigs [3,5,6], little research has shown the acting mechanisms and dose–effects of leptin in embryo implantation. Therefore, this study intended to further assess the effect of different doses of leptin in pig embryo implantation using primary porcine endometrium epithelium cells (PEECs), contributing to the understanding of the mechanisms of reproductive disorders in obesity.

Successful implantation requires four basic conditions: Firstly, a receptive endometrium; β3-integrin, as an adhesion molecule, has been proposed as a marker of uterine receptivity during the window of implantation (WOI) [7,8]; secondly, degradation and remodeling of the extracellular endometrial matrix (ECM) and the expression of adhesion molecules. Matrix metalloproteinases (MMPs) are a family of zinc-dependent proteases that bear the primary responsibility for degrading and rebuilding the bulk of ECM components [9]; thirdly, abundant blood vessels are necessary for supplying nutrients to the embryo. MMPs are crucial for angiogenesis [10] and trophoblast invasion [11] into the endometrium during implantation. Heparin binding EGF-like growth factor (HB-EGF) is also reported to facilitate embryo development [12], and to increase vascular permeability [13]; fourthly, regulation of the immune response and creation of a harmonious fetal–maternal dialogue environment. For the mother, approximately 50% of the embryo is xenobiotic due to paternal genes; as a consequence, implantation has been characterized as an inflammatory-type response. interleukin-1 (IL-1), initially identified as a key mediator of the acute-phase inflammatory response, can modulate communication between the maternal endometrium and embryo [14,15].

In the present study, based on the conditions for successful implantation, we chose β3-integrin, MMP9, HB-EGF, and IL-1β as four representative factors to evaluate leptin’s function in porcine implantation by examining leptin-induced effects on the expression of these factors in PEECs in vitro.

## 2. Materials and Methods

### 2.1. Cell Culture and Treatments

The handling of experimental animals in this study was in accordance with the Guidelines for the Care and Use of Animals in Research enforced by Beijing Municipal Science and Technology Commission. The uterine samples were obtained at an abattoir. One randomly selected uterine horn in each of four Large White sows (day 18 of pregnancy) was ligatured, resected, and washed with sterile PBS (PH 7.4, 8.0 g NaCl, 0.2 g KCl, 1.56 g Na_2_HPO_4_, H_2_O, and 0.2 g KH_2_PO_4_ in 1 L ddH_2_O) supplemented with 100 IU/mL penicillin and 100 μg/mL streptomycin (Gibco, NY, USA) and transported to the laboratory within 1.5 h. The information of pregnant sows and the insemination procedures were as described in our previous studies [4]. In a sterile room, samples were cut longitudinally and washed with fresh ice-cold PBS, and then the embryo was removed from the surface endometrial tissue. Subsequently, endometrial tissue was scraped with a sterile thin glass plate and cut into pieces (about 1 mm^3^). Red blood cell lysing buffer (RBCLB × 10, 41.5 g NH_4_Cl, 5 g KHCO_3_, and 0.15 g EDTA in 500 mL ddH_2_O) was added to the pieces (V:V = 3:1), gently mixed for 10 min, and subsequently centrifuged at 500 g for 10 min to remove red blood cells. The tissue fragments were washed three times with fresh phenol red-free DMEM/F12 medium (Gibco, USA) supplemented with penicillin-streptomycin (100 IU/mL–100 μg/mL) and incubated in phenol red-free DMEM/F12 medium containing 10% (*w*/*v*) fetal bovine serum (FBS, Gibco, NY, USA), 100 IU/mL penicillin, and 100 μg/mL streptomycin at 37 °C in a humidified atmosphere of 5% CO_2_. After the migratory cells from tissues covered 80% of the bottom of the flask (25 cm^2^, Corning), the tissues were removed. Endometrial epithelial cells (EECs) and stroma cells were separated using 0.25% trypsin according to their different sensitivity to trypsin. The homogeneity of cells was evaluated using immunofluorescence staining of cultured cells for the presence of cytokeratin and vimentin as described previously [16]. The purity of EEC cultures was over 90%.

PEECs were cultured for 9–14 days in DMEM/F12 medium containing 10% FBS (Gibco, USA), 100 IU/mL penicillin, and 100 μg/mL streptomycin (complete medium) until they were 80% confluent. The cells obtained from four sows were collected to one pool and washed twice with sterile PBS, then cultured for a further 24 h in the same medium but without FBS (basal medium). Then, cells were washed as described above, and subsequently cultured in basal medium containing leptin (0.00, 0.01, 0.10, 1.00, 10.00 nM) (ProSpec, Rehovot, Israel) for 24 h. The stock solution of leptin (1000 nM) in basal medium was sterilized using a sterilizing filter and afterwards diluted to 0.01, 0.10, 1.00, and 10.00 nM using basal medium when added to cells. When studying signaling pathways, after treatment with basal medium for 24 h, PEECs were acutely pretreated for 30 min with AG490 (40 μM) (Merck, NJ, USA), LY294002 (50 μM) (Santa-cruz, DE, USA), and U0126 (20 μM) (Cell Signaling Technology, MA, USA) to block the JAK2, PI(3)K, and MAPK signaling pathways, respectively. Subsequently, cells were further incubated in the presence or absence of leptin medium (0.01, 1.00, or 10.00 nM) for 24 h. The letpin dose used to treat cells pretreated with blockers was determined according to the results of a previous experiment. Different doses of leptin were used for various target genes based on the effectiveness: β3-integrin, 0.01 nM leptin; MMP9 and HB-EGF, 1.00 nM leptin; and IL-1β, 10 nM leptin. Triplicate wells were run for each treatment and experiments were repeated at least three times with different cell preparations.

### 2.2. RNA Extraction, Reverse Transcription, and Real-Time PCR

After PEECs were treated with leptin, total RNAs were extracted using Trizol reagent (Invitrogen, CA, USA). The RNA concentration, purity, and integrality were verified with NanoDrop (1.8 < A260/A280 < 2.0) and 1% denaturing agarose gels electrophoresis. Qualified RNA samples were treated with DNase to remove DNA, immediately followed by reverse transcription to cDNA using a PrimeScript^TM^ RT reagent Kit (TaKaRa, Dalian, China) in accordance with the manufacturer’s protocol. The expression levels of the mRNA were detected using a real-time quantitative PCR (LightCycler 480 System, Roche) in a reaction volume of 20 μL using a LightCycler 480 SYBR Green RT-PCR Kit (Roche, Basel, Switzerland), including 10 μL of SYBR I Master, 0.5μL of PCR Forward Primer (10 μM), 0.5 μL of PCR Reverse Primer (10 μM), 1 μL of cDNA template, and 8 μL of sterile water. The following cycling conditions were used for all amplifications: 5 min at 95 °C (pre-incubation), 10 s at 95 °C (denaturation), 20 s at 60 °C (annealing), and 30 s at 72 °C (extension). A dissociation stage was added to the PCR procedure to ensure the specific amplification of each primer pair. In addition, a standard curve was generated from a dilution series of cDNA samples to detect the amplified efficiency for each primer pair. The housekeeping gene glyceraldehyde phosphate dehydrogenase (GAPDH) was used as a reference to calibrate RNA levels. The sequences of the primers are shown in Table 1. All experiments were performed with at least three biological replicates and repeated in triplicate. The relative gene expression levels were analyzed using the 2 ^−ΔΔCT^ method.

### 2.3. Protein Extraction and Western Blot Analysis (WB)

Medium was removed from the cultural chambers, then cells were washed twice with sterile PBS, and lysed with ice-cold cell lysis buffer containing a protease inhibitor cocktail, phosphatase inhibitors, and PMSF provided in a total protein extraction kit (KeyGen, Nanjing, China). The cells lysates were in ice for 15 min and centrifuged at 12,000 rpm at 4 °C for 15 min. Supernatants (total protein extract) were kept for protein concentration assays using a BAC protein content assay kit (KeyGen, China). Protein (50 μg) was loaded on SDS-PAGE gels for electrophoresis and transferred to polyvinylidene difluoride (PVDF) membranes, blocked with 5% bovine serum albumin (BSA, ProSpec, Rehovot, Israel) in tris buffered saline tween (TBST) at room temperature for 1 h. The PVDF membranes were immunoblotted with specific primary antibodies (β3-integrin antibody, sc-14009, 1:500; MMP9 antibody, sc-21736, 1:500; HB-EGF antibody, sc-28908, 1:500; IL-1β antibody, AF681, 0.1 μg/mL; p-JAK2 antibody, WL02997, 1:500; p-Akt, CST9271, 1:800; p-Erk antibody, sc-7383, 1:1000) at 4 °C overnight, then washed gently with TBST (three times for 5 min each). Target protein were detected using chemiluminescent substrate (Pierce, USA) and imaged using a flurechemical imaging system (Tanon, Shanghai, China). Quantitative analyses of the Western blot results were performed by densitometry using ImageJ software. The values were normalized to β-actin (β-actin antibody, Biolab, YT633, China) as a loading control.

### 2.4. Statistical Analysis

The data obeyed normal distribution by the Kolmogorov-Smirnov test. Statistical analysis was carried out using the ANOVA function of SAS software (9.1). Turkey’s test was used for multiple comparisons. Data (mean ± standard error) representative results were derived from a minimum of three independent experiments. Values for *p* < 0.05 were considered statistically significant. The model included the main effects of treatments and replicates.

## 3. Results

### 3.1. Leptin Induced the Expression of β3-Integrin, MMP9, HB-EGF, and IL-1β in PEECs

The regulation of leptin on β3-integrin, MMP9, HB-EGF, and IL-1β was initially determined based on the transcriptional expression using RT-PCR (Figure 1A–D). Leptin upregulated β3-integrin, MMP9, HB-EGF, and IL-1β in a dose-dependent manner in PEECs. The 0.01 nM leptin treatment significantly increased mRNA levels of β3-integrin compared with the control treatment (*p* < 0.05), but higher doses of leptin (0.10, 1.00, and 10.00 nM) did not affect the mRNA expression of β3-integrin. PEECs treated with 0.10–10.00 nM leptin had higher MMP9 and HB-EGF mRNA levels than the control group and 0.01 nM leptin group (*p* < 0.05). Among all treatments, MMP9 mRNA peaked at 1nM leptin (*p* < 0.05) as did HB-EGF. With regard to IL-1β, only the highest dose of leptin (10.00 nM) resulted in upregulation (*p* < 0.05).

To further assess the regulatory role of leptin, WB was carried out to determine the expression of β3-integrin, MMP9, HB-EGF, and IL-1β on protein levels in PEECs in response to leptin (Figure 1E–I). All various levels of leptin tested improved β3-integrin, MMP9, and HB-EGF protein expression when contrasted with the control treatment (*p* < 0.05). However, only the two higher levels of leptin (1.00 and 10.00 nM) enhanced IL-1β protein expression (*p* < 0.05). The difference was that β3-integrin, MMP9, and HB-EGF protein expression in PEECs presented their own unique response to increasing leptin. β3-integrin protein did not change with increasing levels of leptin (*p* > 0.05). However, MMP9 protein changed in a parabolic form with increased leptin and peaked when treated with 1 nM leptin (*p* < 0.05). HB-EGF protein rose with the increase in leptin from 0.01 to 0.10 nM (*p* < 0.05) but remained unchanged when treated with higher levels of leptin (0.1 nM-10 nM) (*p* > 0.05).

### 3.2. Phosphorylation of JAK, Akt, and Erk Induced by Leptin Could Be Attenuated by AG490, LY294002, and U0126

The effects of leptin stimulation of Janus kinase 2 (JAK2), *Protein Kinase B* (Akt), and extracellular regulated protein kinases (Erk) phosphorylation in cells treated with AG490, LY294002, and U0126 are shown in Figure 2. As seen from Figure 1, β3-integrin mRNA peaked at 0.01 nM leptin; MMP9 and HE-EGF mRNA peaked at 1.00 nM leptin; and IL-1β mRNA peaked at 10.00 nM leptin. According to this, 0.01, 1.00, and 10.00 nM leptin were chosen for the phosphorylation study. Leptin stimulated the phosphorylation of JAK2, Akt, and Erk in PEECs in a dose-dependent manner (*p* < 0.05). AG490 (40 μM), LY294002 (50 μM), and U0126 (20 μM) blocked the JAK2, PI(3)K, and MAPK signaling pathways respectively by inhibiting the phosphorylation of JAK2, Akt, and Erk (*p* < 0.05). The stimulated effects of leptin on the phosphorylation of JAK2, Akt, and Erk in PEECs receded to different degrees when the signaling pathway was blocked by a matching inhibitor.

### 3.3. JAK2, PI(3)K, and MAPK Signaling Pathways Were Involved in the Leptin Upregulation on β3-Integrin, MMP9, HB-EGF, and IL-1β in PEECs

To determine whether leptin progressed through JAK2, PI(3)K, and MAPK signaling pathways to upregulate β3-integrin, MMP9, HB-EGF, and IL-1β in PEECs, the cells were pretreated for 30 min with AG490, LY294002, and U0126 to block the respective signaling pathways. Subsequently, cells were further incubated in the presence or absence of leptin medium for 24 h. Leptin treatments of cells previously inhibited with blockers showed that the content leptin was unable to upregulate the expression of β3-integrin, MMP9, HB-EGF, and IL-1β mRNA and protein when the JAK2, PI(3)K, or MAPK signaling pathways were blocked (Figure 3, Figure 4 and Figure 5). Those results suggest that leptin regulated the expression of β3-integrin, MMP9, HB-EGF, and IL-1β mRNA and protein through the JAK2, PI(3)K, and MAPK signaling pathways. Blocking any one of the signaling pathways affected the regulation of leptin on the four factors’ expressions, which meant neither pathway was specific for leptin regulation of β3-integrin, MMP9, HB-EGF, and IL-1β mRNA and protein expression. The three pathways may interact with each other or with other pathways during the regulation of embryo implantation by leptin. However, we can still obtain some important information that any factors affecting these three signaling pathways would influence the role of leptin in embryo implantation.

## 4. Discussion

In the present experiment, the relationship between leptin and the four factors (β3-integrin, MMP9, HE-EGF, and IL-1β) in PEECs was evaluated and it was found that leptin improved their expression (Figure 1), which has also been observed in other types of cells, including human/mouse endometrial epithelial cells/tissue [17,18] and bone marrow (BM) progenitor cells [19]. Furthermore, available information reveals that the regulation of leptin for each factor is not independent, and there exists a complex relationship between those factors. It is reported that leptin stimulates IL-1β secretion in both mouse and human endometrial cells [18,20]. IL-1β upregulated the secretion of leptin and expression of leptin and its receptors in human endometrial cells [17]. It is also reported that IL-1 can activate MMP9 [21] in human cytotrophoblasts and trigger the expression of β3-integrin in human endometrial cells [17]. Amazingly, there may be another link in the network; leptin has been shown to increase the gene expression of HB-EGF and extracellular release of HB-EGF in human oesophageal adenocarcinoma cells and this was blocked by MMP9 inhibitor [22]. These phenomena suggest that leptin may link interactions among β3-integrin, MMP9, HB-EGF, and IL-1β and thus participate in porcine implantation directly or indirectly through regulating those adhesion molecules and cytokines.

Angiogenesis is necessary for maternal and fetal gaseous and nutrient exchange [23], and is required for successful implantation. Leptin has been shown to stimulate angiogenesis in epithelial cells [24] and to stimulate corneal neovascularization in rats and leptin-deficient mice [25]. Recently, leptin-integrin cross-talk, as a distinct novel component of the network of angiogenesis, has been proposed in circulating angiogenic cells [26]. Bone marrow MMP9 expression has been shown to be essential for ischemia-induced neovascularization [27]. HB-EGF is also reported to induce a local increase in vascular permeability, which often takes place near the implanting embryo [13]. Our present results showed that leptin upregulated the expression of β3-integrin, MMP9, and HB-EGF in a dose-dependent manner (Figure 1A–C,F–H), all of which were intimately associated with angiogenesis. Taken together, all the results indicated an important role for leptin in porcine implantation through its modulation of angiogenesis.

A receptive endometrium is critical to the arrival of an embryo [8]. β3-integrin is one of the most reliable markers of endometrial receptivity during WOI [7,8] and is also an adhesion molecule allowing embryo adherence and anchoring to the endometrium [28,29,30]. The upregulated efficiency of leptin on β3-integrin (Figure 1A,F) implies that leptin may play a regulative role in uterine receptivity and adhesion of the embryo to the endometrium. The fact that β3-integrin was only upregulated by low levels of leptin (0.01 nM) suggests that obesity, which induces higher levels of leptin, may be detrimental to uterine receptivity.

MMPs are essential in decidualization and embryo invasion by degradation/remodeling of the ECM of the endometrium during implantation. Although absolute decidualization and invasion do not happen in porcine implantation, one of the MMP family members, MMP9, is confirmed to be associated with bovine embryo implantation, which is a non-invasive implantation, albeit some degree of fusion occurs between cells of the surface endometrial epithelium and the trophoblast [31]. Our previous data in vivo also indicates that higher levels of MMP9 protein existed at attachment sites as compared to inter-attachment sites in the endometrium of pregnant pigs [32]. So, it seems reasonable to suggest that MMP9 may also have a regulative role in porcine implantation. The present data show clearly that leptin can improve the expression of MMP9 mRNA and protein in PEECs (Figure 1B,G). These facts suggest that leptin may affect porcine implantation by mediating MMP9 expression. However, an inconsistent result is shown in a recent study, where leptin prevented LPS-induced myometrial remodeling through collagen synthesis stimulation and inhibition of MMP2 and MMP9 in human myometrial explants incubated in vitro for 48 h [33]. However, the latter study sought to assess the role of leptin in obesity-related delivery disorders. Obese people often have unusually high levels of leptin. So, the inhibition of MMP9 by leptin in Wendremaire’s study may be a pathological phenomenon of obese individuals in general. In the present study, it was noteworthy that MMP9 expression in PEECs fluctuated in response to the incremental leptin in a parabolic manner and peaked at 1.00 nM leptin but decreased at 10.00 nM (Figure 1B,G). Based on previous studies [33] and our present data, an inhibitory effect of higher levels of leptin on MMP9 can be speculated. The results also suggested that the expression of MMP9 in obese females may be inhibited, leading to reproductive disorders.

HB-EGF is reported to facilitate embryo development [12], to increase vascular permeability [13], and to attenuate apoptosis of endometrial stromal cells [34]. Leptin is also reported to have a positive effect on embryo development [5], angiogenesis [27], and cell proliferation [35], which suggests that leptin and HB-EGF may act synergistically to regulate porcine implantation. In the present study, the stimulated HB-EGF expression of leptin in PEECs (Figure 1C,H) further confirmed the close interconnection of leptin and HB-EGF in porcine implantation. Appropriate levels of leptin (0.1–10 nM) can increase vascular permeability and accelerate the degradation of stromal cells by upregulating the expression of HB-EGF, which would be beneficial to the exchange of maternal and fetal nutrients and metabolic wastes and embryo invasion into the endometrium.

With regard to IL-1β, evidence shows that IL-1 and leptin are intrinsically related in several cellular functions [36,37]. IL-1β as an inflammatory response cytokine may be involved in immunotolerance at the maternal–placental interface during implantation [15] to help the maternal uterus establish the recognition of pregnancy. Our results showed that the higher levels of leptin (1.00 and 10.00 nM) upregulated the expression of IL-1β but not the lowest leptin (0.01 nM and 0.10 nM) (Figure 1D,I). This suggests that leptin participates in the maternal–placental cross-talk through the regulation of IL-1β-mediated pathways during implantation, and successful maternal–fetal communication may require the mother to increase her fat reserves to appropriate levels, which will induce higher levels of leptin. This may be one reason why mothers gain weight during pregnancy.

According to the current results, it was found that leptin has a dose-dependent effect on the regulation of these embryo implantation-related factors (Figure 1). In other words, too high or too low leptin levels were both not conducive to the successful establishment of embryo implantation. In obese females, there are two extremes of leptin levels. The first is *ob* gene mutation, which causes insufficient leptin production, and the second is resistance to leptin. However, the *ob* gene mutation causing leptin deficiency is rare in the vast majority of obese people, but *ob* gene overexpression inducing hyper serum leptin levels is common [1]. Our results showed that higher levels of leptin were detrimental to downregulation of the mRNA expression of β3-integrin, which regulated the uterine receptivity. So, one of the pregnancy barriers in obese females is likely to be the inability to successfully put the uterus into a receptive state, which was the first critical step of embryo implantation [38]. In healthy women, maternal fat deposition increased as pregnancy went on, implying increasing leptin levels, and the results of this study suggest that moderately increased leptin levels upregulate the expression of MMP, HB-EGF, and especially IL-1, all of which were beneficial to degradation and remodeling of ECM, angiogenesis, and immune recognition of pregnancy.

The signaling pathways implicated in the leptin regulation of implantation are complex. JAK2, PI(3)K, and MAPK signaling pathways are activated by leptin in several cell types [39,40]. Based on this point, JAK2, PI(3)K, and MAPK signaling pathways were chosen to assess whether they were linked to the regulation of leptin on β3-integrin, MMP9, HB-EGF, and IL-1β in PEECs. The results indicated that leptin can activate the phosphorylation of JAK2, Akt, and Erk, and AG490, LY294002, and U0126 all mitigated the upregulative effect of leptin on β3-integrin, MMP9, HB-EGF, and IL-1β in PEECs by inhibiting the phosphorylation of JAK2, Akt, and Erk. The results suggested that JAK2, PI(3)K, and MAPK are involved in the upregulation of β3-integrin, MMP9, HB-EGF, and IL-1β induced by leptin in PEECs. JAK2, PI(3)K, and MAPK were pathways not specific for obesity and reproduction. IRS-P(3)K and MAPK are the main pathways of insulin signal transduction [41], which is closely connected with obesity. Leptin is able to activate IRS1/2-associated PI(3)K [42]. PI3K activation leads to activation of the main downstream target, Akt. Akt is a cytoplasmic serine kinase that is important in regulating cell growth, differentiation, adhesion, and inflammatory reactions [43]. The phosphorylated JAK2 (p-JAK2), through Ras protein and Raf kinase, can also activate the MAPKKK/MEK/MAPK pathway, which is called the JAK2-Ras-Raf-MEK-MAPK signaling pathway. MAPK seems to have various roles, including regulating cell growth, differentiation, and apoptosis; facilitating vascular endothelial cell proliferation and neovascularization; and mediating inflammatory responses. In a similar way, AG490, as the inhibitor of JAK2, can also block the JAK2-IRS-PI(3)K-Akt and JAK2-Ras-Raf-MEK-MAPK signaling pathways when it blocks the JAK2/STAT3 signaling pathway. Interestingly, the leptin-JAK2-STAT3 pathway is currently thought to only mediate energy balance in the body but not to affect reproduction [44]. The literature now implies that the dominant roles in the leptin regulation of β3-integrin, MMP9, HB-EGF, and IL-1β expression in PEECs may be played by the JAK2-IRS-PI(3)K-Akt and JAK2-Ras-Raf-MEK-MAPK signaling pathways; however, further studies are still necessary.

## 5. Conclusions

In conclusion, leptin regulates β3-integrin, MMP9, HB-EGF, and IL-1β expression in a dose-dependent manner. According to the present results, JAK2, PI(3)K, and MAPK may participate in the process of leptin-mediated porcine implantation. The findings will contribute to further understanding of the mechanisms of reproductive disorders in obesity.

## Figures and Tables

**Figure 1 ijerph-17-06508-f001:**
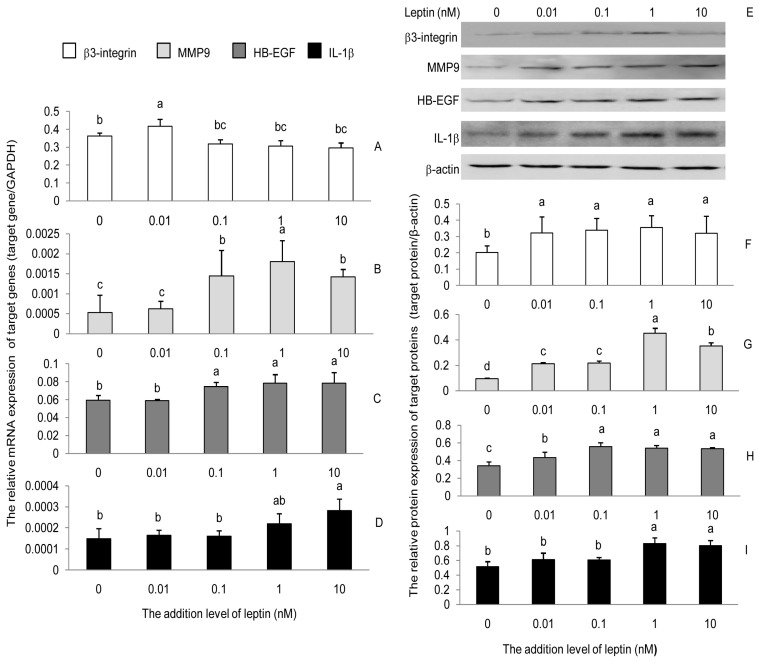
Expression of β3-integrin, MMP9, HE-EGF, and IL-1β mRNA and proteins in primary porcine endometrium epithelial cells (PEECs) in response to leptin. (**A**–**D**): The relative mRNA expression of β3-integrin, MMP9, HE-EGF, and IL-1β in response to leptin determined by RT-PCR. GAPDH was used as the internal control. (**E**–**I**): The relative protein expression of β3-integrin, MMP9, HE-EGF, and IL-1β in response to leptin determined by Western blot analysis (WB). β-actin was chosen as the loading control to normalize the target protein, and densitometric analysis of bands was carried out using ImageJ software. Columns with a different lowercase letter represent statistically significant differences (*p* < 0.05). Data (means ± standard error) are representative results derived from a minimum of three independent experiments.

**Figure 2 ijerph-17-06508-f002:**
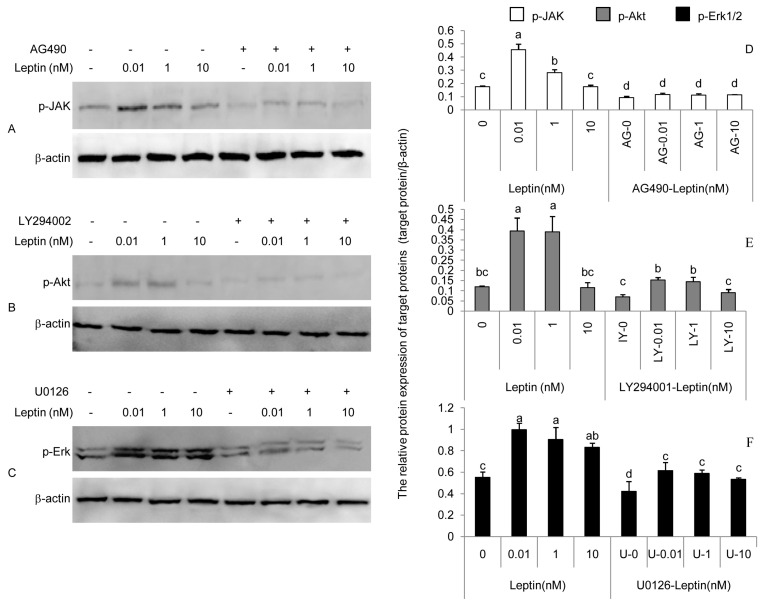
Effects of leptin and inhibitors on the protein expressions of p-JAK2, p-Akt, and p-Erk1/2 in primary porcine endometrium epithelial cells (PEECs) in response to leptin and inhibitors. (**A**–**C**): Western blot analysis (WB) pictures of p-JAK2, p-Akt, and p-Erk1/2. (**D**–**F**): Quantification of the WB pictures using densitometry. β-actin was chosen as the loading control to normalize the target protein and densitometric analysis of bands was carried out with Image J software. The p-Erk protein level was expressed using the total densitometric value of the two bands. Columns with a different lowercase letter represent statistically significant differences (*p* < 0.05). Data (means ± standard error) are representative results derived from a minimum of three independent experiments.

**Figure 3 ijerph-17-06508-f003:**
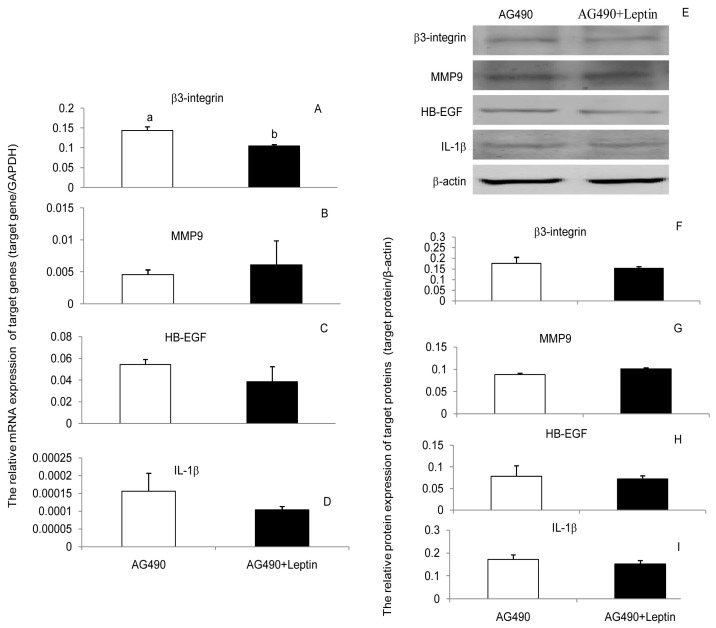
Effect of leptin and AG490 (inhibitor of JAK2 expression) on the expression of β3-integrin, MMP9, HE-EGF, and IL-1β mRNA and proteins in primary porcine endometrium epithelial cells (PEECs). (**A**–**D**): The relative mRNA level of β3-integrin, MMP9, HE-EGF, and IL-1β in response to leptin and AG490 determined by RT-PCR. GAPDH was used as the internal control. (**E**–**I**): The relative protein expression of β3-integrin, MMP9, HE-EGF, and IL-1β in response to leptin and AG490 determined by wester blot analysis (WB). β-actin was chosen as the loading control to normalize the target protein and densitometric analysis of bands was carried out with ImageJ software. Columns with a different lowercase letter represent statistically significant differences (*p* < 0.05). Data (means ± standard error) are representative results derived from a minimum of three independent experiments.

**Figure 4 ijerph-17-06508-f004:**
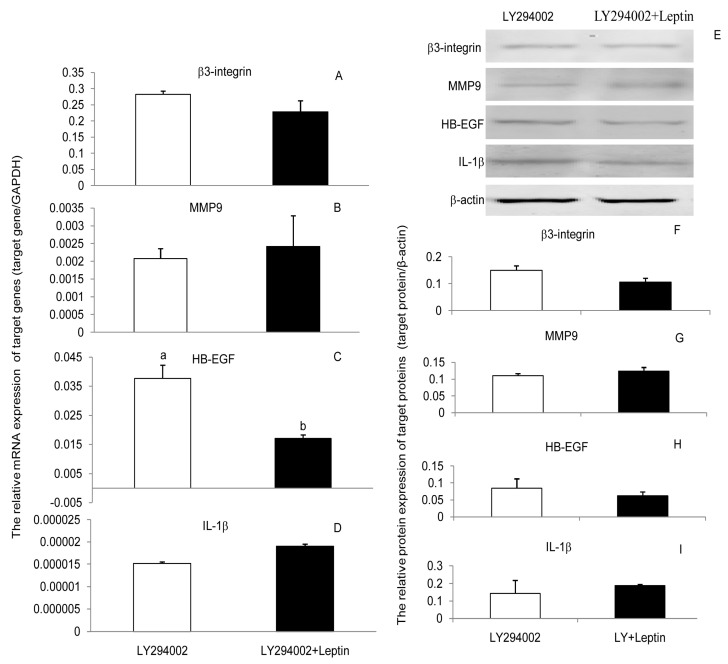
Effect of leptin and LY294002 (inhibitor of PI(3)K) on the expression of β3-integrin, MMP9, HE-EGF, and IL-1β mRNA and proteins in primary porcine endometrium epithelial cells (PEECs). (**A**–**D**): The relative mRNA level of β3-integrin, MMP9, HE-EGF, and IL-1β in response to leptin and LY294002 determined by RT-PCR. GAPDH was used as the internal control. (**E**–**I**): The relative protein expression of β3-integrin, MMP9, HE-EGF, and IL-1β in response to leptin and LY294002 determined by Western blot analysis (WB). β-actin was chosen as the loading control to normalize the target protein and densitometric analysis of bands was carried out using ImageJ software. Columns with a different lowercase letter represent statistically significant differences (*p* < 0.05). Data (means ± standard error) are representative results derived from a minimum of three independent experiments.

**Figure 5 ijerph-17-06508-f005:**
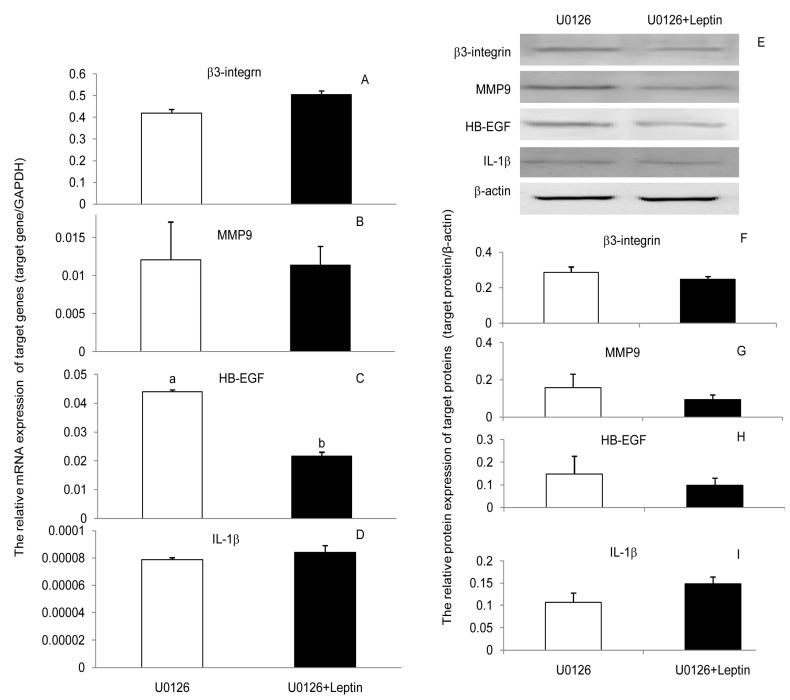
Effect leptin and U0126 (inhibitor of MAPK) on the expressions of β3-integrin, MMP9, HE-EGF, and IL-1β mRNA and proteins in primary porcine endometrium epithelial cells (PEECs). (**A**–**D**): The relative mRNA level of β3-integrin, MMP9, HE-EGF, and IL-1β in response to leptin and U0126 as determined by RT-PCR. GAPDH was used as the internal control. (**E**–**I**): The relative protein expression of β3-integrin, MMP9, HE-EGF, and IL-1β in response to leptin and U0126 as determined by Western blot analysis (WB). β-actin was chosen as the loading control to normalize the target protein, and densitometric analysis of bands was carried out using ImageJ software. Columns with a different lowercase letter represent statistically significant differences (*p* < 0.05). Data (means ± standard error) are representative results derived from a minimum of three independent experiments.

**Table 1 ijerph-17-06508-t001:** The sequences of primers.

Gene	Gene No.	Primer Sequence	Length (bp)
β3-integrin	NM_214002	F: 5′-CTTCTATTTGGGAGTGAG-3′	167
		R: 5′-AGGTGATGCTGGTCTAA -3′	
MMP9	NM_001038004	F: 5′-CACGCATTGGGCTTAGAT-3′	130
		R: 5′-TAGGGCGAGAACCATACA-3′	
HB-EGF	NM_214299	F:5′-AAAGAAGAAAGGCAAAGGG-3′	242
		R: 5′-GACAGACGGACGACAGCA-3′	
IL-1β	NM_001005149	F:5′-AAGTGATGGCTAACAATG-3′	266
		R: 5′-TTCTTCAAAGACGGATG-3′	
GAPDH	NM_001206359.1	F: 5′-GTCCACTGGTGTCTTCACGA-3′	145
		R: 5′-GCTGACGATCTTGAGGGAGT-3′

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
