# Peer review of "Leptin Upregulates the Expression of β3-Integrin, MMP9, HB-EGF, and IL-1β in Primary Porcine Endometrium Epithelial Cells In Vitro"

_ijerph, 2020, doi:10.3390/ijerph17186508_

Round 1

Reviewer 1 Report

Implantation of embryos is a complex process. Embryonic implantation and establishment of successful pregnancy include dynamic process of interactions between the embryo and a receptive maternal endometrium. A variety of molecules and complex interactions were identified as potential mediators of embryo-uterine interactions during implantation. For the last several decades, the list of “implantation” regulators is expanding. Initially, the embryo has to attach its placental cells to the surface cells of the endometrium. This is a process that is mediated by a complex of proteins expressed both on the surface of the embryo and on the surface of the endometrium. The secretion of the embryonic MMP-9 is closely related to

the quality of the embryo and embryo implantation. Role of HB-EGF as the earliest indicator of embryonic signals is unique. There are described major roles of b3-integrin and IL-1b as regulators of local cellular interactions during implantation of embryo.

There is additional problem for successfully pregnancy in obese woman because of leptin level. It is known that leptin is a 16-kDa multifunctional protein. Understanding mechanism how leptin interact with b3-integrin, MMP9, HB-EGF, and IL-1b14 and how influence to theirs levels may be crucial for endometrial receptivity in obese woman.

This manuscript represents additional effort as contribution for understanding mechanism of regulation uterine receptivity and embryo adhesions to endometrium.

Author Response

We appreciated highly the views of reviewer 1 and  thank the reviewer for the pertinent comments.

Reviewer 2 Report

It is a robust work that involves the assessment of the hormone Leptin, at the levels of β3-integrin, MMP9, HB-EGF and IL-1β, which are involved in four stages of embryonic implantation.
The study is well delimited, where the hypothesis is answered with the results found, however, it is a low innovation study. The role of the hormone has been evaluated over decades, in several species, including humans.
I believe that the results are supported by molecular tests of PCR and WB, however, data from immunohistochemistry or immunocytochemistry would also add value and corroborate these results. If you have such data, don't hesitate to add it.
Regarding statistics, I think you only need to describe the figures in their representations a, b, c better on the bars in order to facilitate the interpretation of the graphs.

Author Response

We are glad that the reviewer has affirmed the value of this paper and we would also like to thank the reviewers for their comments.

  • 1) We admired the reviewer's broad knowledge and insight in this field. It is easy to infer the important role of leptin in reproduction based on existing research progress. Soit's really not a very innovative study. But the detailed effects of leptin on β3-integrin, MMP9, HB-EGF and IL-1β during embryonic implantation have not be shown specifically. We presented some basic data matter-of-factly, hoping to contribute to better understanding of leptin's reproductive function.
  • 2) Data of immunohistochemistry or immunocytochemistry would really add value of those results. We have done immunohistochemistry, but the data have been published (Wang et al., 2014). But we have no data of So I'm really sorry that this part of data cannot be added at present. However, we think the current data are telling.

Hongfang Wang, Jinlian Fu, Aiguo Wang. Expression of obesity gene and obesity gene long form receptor in endometrium of Yorkshire sows during embryo implantation. Mol Biol Rep (2014) 41:1597–1606.

  • 3) About the comment “Regarding statistics, I think you only need to describe the figures in their representations a, b, c better on the bars in order to facilitate the interpretation of the graphs.”

I am sorry that I did not understand what the reviewer meant. I annotated the lowercase letter on the bars as following.

“Columns with a different lowercase letter represent statistically significant differences (P < 0.05)”

I think this annotation can help the reader understand the figures.

Reviewer 3 Report

This study deals with the mechanisms of reproductive disorders in obesity by measuring of mRNA and protein expression levels of beta3-integrin, MMP9, HB-EGF, and IL-1beta. This study attempts to provide an important contribution to implantation of porcine. While the contents of this experiments are sound and present well, I have concerns about the results and discussion. There are some improvements that should be made before publication, which I believe will improve the readability of the paper.

Major points

  • Confirm the experimental design of the measurement of mRNA expression levels. The primers design of beta3-integrin and IL-1beta is wrong.
  • Overall, explanation and interpretation of results is not enough. Particularly, there are almost no explanation of Fig 3-5.
  • The result and discussion part must be improved with more explanations and discussions. Please cite the results and figures.
  • The background information (including the purpose of study) and the content of discussion is not consistent. The signaling pathway is not specific for obesity and this relationship is not discussed well. Moreover, appearance of insulin pathway is abrupt.

Minor points

Lines 67- 72: Capital and small letters are mixed, such as NaCL, NaCl, and KCL.

Overall: Please provide the company information. Unify the style. Location is missing.

Line 67: Between PH and 7.4, equal is removed.

Line 88: How authors prepared leptin?

Line 94: “RAN extraction” should be “RNA extraction”.

Line 97: Provide the transcriptase details. Superscript?

Table 1: As mentioned above, confirm the primer sequences of beta3-integrin and IL-1beta. Moreover, the length of GAPDH should be 145 bp.

qPCR and WB: Why author use the different genes for internal control? GAPDH and actin.

Line 120: How authors prepared antibody of beta-actin?

Overall: Regarding of statistical differences of P<0.05, the letter of “P” should express by the italic and small letters.

Lines 133-134: This sentence is difficult to follow. Please rephrase.

Figures 2H and 2I: Please explain why protein expression elevated before mRNA expression increased.

Section 3.3: As mentioned above, results were not expressed.

Figures 3B and 3C: Wrong letters (statistical differences) are put.

Lines 221-225: Authors should not provide cancer cases.

Line 298: Reference 41 is not appropriate. This situation is not anorexia.

Line 428: After Littell, R.D should be semi-colon.

Line 431: Regarding Morton, G.J, remove the semi-colon before J

Author Response

We are glad that the reviewer has affirmed the value of this paper and we would also like to thank the reviewers for the detailed comments which would be of great value in improving the quality of the manuscript We have revised the questions raised by reviewer point by point as following.

Reviewer 4 Report

In the MS under analysis, entitled "Leptin Up-Regulated the Expression of b3-integrin, 2 MMP9, HB-EGF, and IL-1b in porcine Endometrium Epithelial Cells in Vitro" Hongfang Wang and colleagues report the effects of Leptin on cultured 18-days pregnancy endometrial epithelial cells 1) over the expression of four molecules (identified in the MS title) in cell cultures inhibited or not with AG490, LY294002, and 158 U0126; and 2) on the phosphorylation of three different kinase proteins. With this study, the authors intend to unveil a putative mechanism of Leptin in (in)fertility.
This is an interesting study that may benefit those working in the field of Reproduction, including men. However, the authors should provide additional information on particular issues before being considered suitable for publication. Below I identify some of my primary concerns. Others were included in the commented copy of the MS attached to this review.

  1. In M&M, the authors describe the collection od uterine samples from 18-day pregnant uteri. However, embryos begin to attach to the uterus on days 13–14 of pregnancy in the pig. When scraping the endometrial surface, endometrial epithelial cells would be obtained and cells from the trophoblast. How did the authors distinguish the two main populations? Authors describe the methods used to separate epithelial cells from the stroma, but not form trophoblastic cells
    In Discussion, the authors mention that the pathways studied in this experiment are associated and that a complex interplay exists between the target molecules and leptin regulation. This raises one question: If this is a fact, then how did the authors separated the direct effects of Leptin from those associated with the other molecules (either additive, synergic, or antagonistic)?
  2. The reasons to select the molecules in the study and the pathways addressed in their research, along with their relation to pig implantation should be included in the introductory section
  3. The M&M section needs the attention of the authors because there is information amiss that is important for the transparency of the study (for detail, please see the comments in the attached copy of the MS)
  4. In the results section, some sentences need to be reworded for clarification. Also, some of the information there respects M&M and should be moved into that section
  5. The Discussion needs to be improved. Frequently, the content is too superficial [To talk about the effects of the target molecules on cell proliferation, neovascularization or apoptosis and inflammation in a general way is insufficient; if the authors studied their effects at implantation, then the associations ought to be on implantation-related events], and some of the comparisons made with other species are incongruent and do not take in consideration the particularities in the morphology or the physiology of the species. For example, on lines 257/8, authors say that bovine and pig embryo implantation are similar non-invasive placenta. They ignored that, in the cow, the superficial invasion of the trophoblast cells to fuse and form the Binucleate cells. The placenta is of the syndesmochorial type. This phenomenon is not described in pigs, which present an epitheliochorial placenta.

Author Response

We are glad that the reviewer has affirmed the value of this paper and we would also like to thank the reviewers for the detailed comments which would be of great value in improving the quality of the manuscript. We have revised the questions raised by reviewer point by point as following.

Round 2

Reviewer 3 Report

The manuscript has been revised well. I think this manuscript will be acceptable after some corrections have been done.

  • Line 43: Need a space between reference 3, and 5.
  • Line 96: Check the spelling. Merk should be Merck.
  • Line 110: Regarding PrimeScriptTMRT, TM (trademark) is expressed with the superscript letter. Otherwise remove “TM”.
  • Line 126: Remove the sequence-like letters “ TTCTT….”
  • Fig 3B and 3C, letters (statistical differences) should be removed. The authors replied “There were not significant statistically.”
  • Lines 281, 301, 309, 331, Remove the space between figure numbers and letters. For example, Figure 1A. Regarding line 331, an extra space was found before parenthesis.

Author Response

Thank the reviewer for the positive comments and new comments. We have made modifications one by one as follows. To distinguish it from the first modification, the second modification is covered with yellow background in the revised manuscript.

Reviewer 4 Report

In their revised MS, the authors tried to address most of my comments, which I value.

Still there are some issues remaining that need the authors attention, manely:

  1. auhtors should explain why did they choose different control/reference gene for each assay it may bias/compromise data analysis. For that reason, the comparison of WB and PCR data can not be made directly. And this issue fragilises the MS soundness
  2. concerning the statistical analysis, data follows or do not follow the normal distribution. "approximatly" is not accetable. Authors should clarify the issue as it determines the suitability of the statistical analysis 
  3. the p-Erk blot shows two bands (precursor and activated molecules possibly). How was the measurement/quantification made? the mean value of the two bands? or only one band was analysed? please clarify and include this clarification in M&M
  4. lines 209-217 - This is discussion. Move the sentence accordingly
  5. concerning figures 3.B and 3.C, authors should explain how one group is simultaneously similar and different from the other, when only two groups were used?
  6. in the Discussion section avoid to mention the MS figures
  7. lines 325-326 -- Auhtors did not test the success of implantation, so these kind of speculative sentences should be avoid
  8. lines 336-337 (and lunes above) the current study focused on pregnancy day 18. it is speculative to conclude that the cellular conditions would be maintained through pregancy and that cells (with a different metabolic age and a different role - trophoblast cells evolute during pregnancy: they are not the same at the begining and at the end) would responde in the same way throughout pregnancy.
  9. lines 366-367 - this is not a conclusion to be drawn from the current study. authors should reconstruct this section.

please address to the commented file in attcah for minor issues

Best regards

Author Response

I really appreciate the academic opinions and attitudes of the reviewer. The process of revising a manuscript is more like an academic exchange with a reviewer, which enabled me to learn a lot and greatly improved the quality of the manuscript. I made the corrections and replies one-by-one as follows. To distinguish it from the first modification, the second modification is covered with yellow background in the revised manuscript.
